# Bronchoalveolar Lavage Cell Count and Lymphocytosis Are the Important Discriminators between Fibrotic Hypersensitivity Pneumonitis and Idiopathic Pulmonary Fibrosis

**DOI:** 10.3390/diagnostics13050935

**Published:** 2023-03-01

**Authors:** Małgorzata Sobiecka, Monika Szturmowicz, Katarzyna B. Lewandowska, Inga Barańska, Katarzyna Zimna, Ewa Łyżwa, Małgorzata Dybowska, Renata Langfort, Piotr Radwan-Röhrenschef, Adriana Roży, Witold Z. Tomkowski

**Affiliations:** 11st Department of Lung Diseases, National Tuberculosis and Lung Diseases Research Institute, Plocka 26, 01-138 Warsaw, Poland; 2Department of Radiology, National Tuberculosis and Lung Diseases Research Institute, Plocka 26, 01-138 Warsaw, Poland; 3Department of Pathology, National Tuberculosis and Lung Diseases Research Institute, Plocka 26, 01-138 Warsaw, Poland; 4Department of Genetics and Clinical Immunology, National Tuberculosis and Lung Diseases Research Institute, Plocka 26, 01-138 Warsaw, Poland

**Keywords:** bronchoalveolar lavage lymphocytosis, bronchoalveolar lavage cell count, fibrotic hypersensitivity pneumonitis, idiopathic pulmonary fibrosis, diagnosis

## Abstract

Background: Fibrotic hypersensitivity pneumonitis (fHP) shares many features with other fibrotic interstitial lung diseases (ILD), and as a result it can be misdiagnosed as idiopathic pulmonary fibrosis (IPF). We aimed to determine the value of bronchoalveolar lavage (BAL) total cell count (TCC) and lymphocytosis in distinguishing fHP and IPF and to evaluate the best cut-off points discriminating these two fibrotic ILD. Methods: A retrospective cohort study of fHP and IPF patients diagnosed between 2005 and 2018 was conducted. Logistic regression was used to evaluate the diagnostic utility of clinical parameters in differentiating between fHP and IPF. Based on the ROC analysis, BAL parameters were evaluated for their diagnostic performance, and optimal diagnostic cut-offs were established. Results: A total of 136 patients (65 fHP and 71 IPF) were included (mean age 54.97 ± 10.87 vs. 64.00 ± 7.18 years, respectively). BAL TCC and the percentage of lymphocytes were significantly higher in fHP compared to IPF (*p* < 0.001). BAL lymphocytosis >30% was found in 60% of fHP patients and none of the patients with IPF. The logistic regression revealed that younger age, never smoker status, identified exposure, lower FEV_1_, higher BAL TCC and higher BAL lymphocytosis increased the probability of fibrotic HP diagnosis. The lymphocytosis >20% increased by 25 times the odds of fibrotic HP diagnosis. The optimal cut-off values to differentiate fibrotic HP from IPF were 15 × 10^6^ for TCC and 21% for BAL lymphocytosis with AUC 0.69 and 0.84, respectively. Conclusions: Increased cellularity and lymphocytosis in BAL persist despite lung fibrosis in HP patients and may be used as important discriminators between IPF and fHP.

## 1. Introduction

Recently, the American Thoracic Society (ATS), Japanese Respiratory Society (JRS) and Association Latinoamericana del Torax (ALAT), as well as the American College of Chest Physicians (ACCP) have independently published practical clinical guidelines for the diagnosis of hypersensitivity pneumonitis (HP) [1,2]. Both documents provide algorithms to establish the diagnosis in a patient with interstitial lung disease (ILD) suspected to have HP based on a combination of specific features grouped into three domains: 1. exposure identification, 2. high-resolution computed tomography (HRCT) pattern, and 3. bronchoalveolar lavage (BAL) lymphocytosis. Based on the ACCP guidelines, a confident diagnosis of HP can be made in a patient who has an inciting antigen identified and typical pattern on a HRCT scan, without the need for invasive procedures [1]. According to the ATS/JRS/ALAT guidelines, a highly confident diagnosis of HP additionally requires the presence of lymphocytosis in the bronchoalveolar lavage fluid (BALF) equal to 30% or greater [2]. 

HP is a complex, immune-mediated interstitial lung disease caused by the repeated inhalation of organic dust in susceptible individuals [3]. The disorder shares the features of other acute/inflammatory or chronic/fibrotic pulmonary diseases. As a result, fibrotic hypersensitivity pneumonitis (fHP) can be misdiagnosed as idiopathic pulmonary fibrosis (IPF), especially when the inciting antigen has not been identified, despite a thorough history [4]. Until recently, HP patients have been classified according to symptom chronicity in acute and chronic forms [5], but both of the last published guidelines propose that HP be simply classified as fibrotic or nonfibrotic based on the presence or absence of fibrosis on the HRCT of the chest and/or histopathological analysis [1,2]. Based on new data, stratification according to the presence of fibrosis seems more in line with the prognosis and may have important diagnostic and therapeutic implications [1]. The most frequent and challenging diagnostic dilemma encountered by ILD-experienced clinicians is the differentiation of fibrotic HP from IPF due to overlapping symptoms, HRCT pattern, and histological findings. The discrimination between fibrotic HP and IPF represents a frequent diagnostic challenge even in the best tertiary lung disease centres. An integrated approach to the assessment of clinical features, radiological patterns, and bronchoalveolar lavage fluid and/or histopathological findings (as appropriate) should be applied according to the recent ATS/European Respiratory Society (ERS)/JRS/ALAT guidelines for IPF [6] and ATS/JRS/ALAT [2] or ACCP guidelines for fHP [1]. A thorough medical history of exposure and evaluation of HRCT images by an ILD-experienced radiologist are crucial in the differential diagnosis of fHP and IPF. However, in many cases of fHP, an exposure is not identified (up to approximately 50% of cases). In addition, lung fibrosis on HRCT, not accompanied by the radiologic features of active HP, such as the centrilobular nodules or the areas of grand-glass opacities, may be difficult to differentiate from IPF. On such occasions, the coexistence of fibrosis and air trapping and the distribution of the lesions (with no basal predominance of the lesion) may determine the suspicion of fHP [2]. Thus, only fair agreement on a diagnosis of HP across multidisciplinary discussion groups has been demonstrated, while for other ILDs, the agreement was good [7]. 

BALF analysis is frequently used in patients with newly identified ILD, as a low-risk procedure that can narrow the differential diagnosis, and in some cases, may eliminate the need for a lung biopsy [8]. Increased cellularity with lymphocytosis is associated with HP [9,10] and the threshold of 30% for lymphocytes in BAL has been proposed to be reasonable in distinguishing HP from non-HP ILD [2]. Re-evaluation of patients with different fibrotic ILD driven by BAL results as a complementary tool can lead to a change in diagnosis in some cases [4,11]. 

However, the role of BALF cell count and lymphocytosis and their discriminative performance in the distinction between fHP and IPF needs to be clarified. We aimed to determine the frequency and magnitude of BALF lymphocytosis in patients with fHP and IPF, and to evaluate the best cut-off points for total cell count and percentage of lymphocytes discriminating these two fibrotic ILDs. We hypothesized that BALF lymphocytosis and total cell count may have a valuable complementary role when differentiating between fHP and IPF.

## 2. Materials and Methods

### 2.1. Ethical Approval

The study was approved by the Bioethical Committee at the National Tuberculosis and Lung Diseases Research Institute (approval No. KB-14/2019 and KB-5/2022) and conducted in accordance with the Declaration of Helsinki. Patients’ consent was waived by the Bioethical Committee because of the study’s retrospective nature. All personal data were anonymized. The publication does not include any data or features enabling the identification of any individual patient in the analysis.

### 2.2. Study Population

Patients with fibrotic hypersensitivity pneumonitis and idiopathic pulmonary fibrosis were identified retrospectively from a cohort of consecutive patients diagnosed with different ILDs in our tertiary referral centre between 2 January 2005, and 31 December 2018, and their initial diagnoses were re-evaluated by our multidisciplinary team. The diagnosis of fHP was based on the current ATS/JRS/ALAT guidelines [2] through multidisciplinary discussion (MDD), following the integration of clinical data, exposure history, HRCT pattern, bronchoalveolar lavage, and histology, when available. IPF was diagnosed according to the 2018 ATS/ERS/JRS/ALAT statement [6] and its update from 2022 [12]. Only patients who underwent bronchoalveolar lavage in the diagnostic process were eligible for the study. 

### 2.3. Data Collection and Procedures

Data regarding age, gender, smoker status, history of exposure, HP precipitin serology, pulmonary function test, 6 min walk test distance and desaturation, and bronchoalveolar lavage fluid results were extracted from the hospital database. 

A detailed description of the diagnostic procedures used by our group has been published previously [13,14].

Spirometry and the whole body plethysmography were performed as routine measures in all patients with an integrated measuring device Master Screen Body/Diffusion by Jaeger (Germany 2002), following the ERS/ATS recommendations [15,16] and reported as percentages of predictive values according to the ERS reference equations [17]. The transfer factor of the lungs for carbon monoxide (TL,co) was measured with a single breath method, with helium gas as a marker. The results were presented as a percentage of the predicted values with a correction to haemoglobin concentration [18]. The six-minute walk test was performed on a corridor in accordance with ATS guidelines [19] and the distance and desaturation at the sixth minute were noted. 

Bronchoalveolar lavage was performed for diagnostic purposes according to the ATS recommendations [8]. The bronchoscope was placed in the wedge position in a subsegmental bronchus of the middle lobe. Then, warmed up to 37 Celsius degrees 0.9% saline was instilled through a working channel in 20 mL aliquots, up to a maximum of 200 mL. The recovered BAL fluid (at least 50% of infused volume) was filtered through sterile gauze, and centrifuged (4 °C, 400× *g*, 15 min). Cell viability was assessed by trypan blue exclusion. The total cell count was counted using a Bürker chamber and differential cell count was evaluated in light microscopy on May–Grunwald–Giemsa stained slides by counting a minimum of 600 cells [14]. None of the patients had received glucocorticosteroids or immunosuppressants prior to the BAL being performed. 

### 2.4. Statistical Analysis

Statistical analyses were performed in Stata 15.1. Two-sided α = 0.05 was used to determine statistical significance. The one-way ANOVA and chi-squared test were used to assess differences in descriptive statistics in continuous and categorical variables, respectively. Logistic regression was used to evaluate the potential diagnostic utility of clinical findings in differentiating fHP and IPF, adjusting for age, sex, and smoking status (never-smoker versus ever-smoker). BALF parameters with evidence of discriminatory performance were evaluated for their diagnostic performance based on the receiver operating characteristic curve (ROC), and positive and negative predictive values. Optimal diagnostic cut-offs were established based on maximising the product of sensitivity and specificity and they were rounded to the nearest integer [20].

As a sensitivity analysis, multivariable logistic regression models were further adjusted for estimated pack-years of smoking (*n* participants = 127). *p* ≤ 0.05 was defined as a significant difference. 

## 3. Results

### 3.1. Baseline Characteristics of the Study Group

A total of 136 patients (65 with fibrotic hypersensitivity pneumonitis and 71 with idiopathic pulmonary fibrosis) were included in the study. The baseline clinical characteristics and pulmonary function test findings of the study population are summarized in Table 1 and Table 2, respectively. The patients in the fHP group were significantly younger, more often female and never-smokers than the IPF patients. An exposure history to various antigens was recognized in 72.3% of fHP patients and it was dominated by avian antigens. There were no significant differences in the results of six-minute walk distance, desaturation and pulmonary function tests between fHP and IPF patients, except for FEV1 and FVC, which were lower in the patients with fHP.

### 3.2. BALF Cell Analysis

BALF cell analysis results and distribution of lymphocytosis by diagnosis are presented in Table 3. The total cell count and the percentage of lymphocytes were significantly higher in the fHP group compared to the IPF group (*p* < 0.001) (Figure 1a,b). Among patients with fHP BALF lymphocytosis equal to or greater than 20% was observed in 75% of cases, while only in 10% of patients with IPF. In addition, BALF lymphocytosis above 30% was found in 60% of fHP patients, and none of the IPF patients. 

### 3.3. Predictors of Fibrotic HP Diagnosis

A logistic regression analysis adjusted for age, gender, and smoking status was performed to determine predictors of fHP diagnosis. The analysis revealed that younger age, never smoker status, identified exposure, lower FEV1 (in % predicted value), higher BALF cell count and lymphocytosis significantly increased the odds of fHP diagnosis rather than IPF. The BALF lymphocytosis exceeding 20% increased the odds of being classified as fHP rather than IPF by 25-fold (Table 4). The sensitivity analysis using the multivariable logistic regression models further adjusted for estimated pack-years, provided similar findings (Appendix A).

The optimal cut-off values to differentiate fibrotic HP from IPF were 15 × 10^6^ for BALF cell count and 21% for lymphocytosis. With the use of calculated cut-off values, the sensitivity and specificity of BALF lymphocytosis for the recognition of fHP were 75% and 93%, respectively, and the corresponding values for BALF cell count, were 80% and 59%, respectively (Table 5).

The receiver operating characteristic (ROC) curves illustrating the diagnostic utility of BALF total cell count and percentage of lymphocytes in differentiating fibrotic HP from IPF are shown on Figure 2 and Figure 3, respectively. 

## 4. Discussion

There are no randomised controlled trials or controlled observational studies evaluating the BALF lymphocyte percentage as a diagnostic test for fHP. On the other hand, many authors present the difficulties in the differential diagnosis between fHP and IPF in clinical practise. This is caused by a relatively large population of fHP patients, in whom the HRCT pattern is not typical of HP, or even presents the usual interstitial pneumonia (UIP)-like features [4,11,21]. On such occasions, the MDD experts decide to perform invasive diagnostic procedures to ascertain the diagnosis [1]. 

In our retrospective study, we found that not only lymphocytosis, but also total cell count in BALF is important in distinguishing fHP from IPF, with an AUC at 0.90 for lymphocytosis and at 0.71 for cell count. We also showed that the BALF lymphocytosis exceeding the 20% increased by twenty-five times the probability of being classified as fHP rather than IPF. In addition, in our study, it was demonstrated that a lymphocytosis equal to or higher than 21% and a total cell count of 15 × 10^6^ in BALF would be appropriate cut-offs to consider fHP as a diagnosis, with a sensitivity of 75% and 80%, and with a specificity of 93% and 59%, respectively, for discriminating fHP from IPF.

In the diagnostic evaluation of patients with fibrotic ILD, the distinction between fHP and IPF can be challenging even for experienced clinicians, in particular when the exposure to the antigen has been hidden or forgotten [4,10]. Nevertheless, this differentiation is crucial for disease management and prognostication, especially in the era of antifibrotic treatment. The avoidance of further exposure to the identified antigen and consideration of immunosuppressive therapy remain essential in the management of patients with fHP, while the prompt initiation of antifibrotic treatment is essential in IPF [22,23,24]. 

The primary goal in the diagnosis of ILD is to make a confident diagnosis using the least invasive approach, considering that these patients are often elderly and have some comorbidities. Hence, the usefulness of BALF cellular analysis, including lymphocytosis in the differentiation of fHP from IPF and other fibrotic ILDs, as a minimally invasive method, is a subject of debate in the literature. 

Two recent systemic reviews and meta-analyses have pooled the data on the value of BALF lymphocytosis in the diagnosis of HP and found higher lymphocyte percentages in chronic/fibrotic HP compared to IPF [25,26]. In a meta-analysis of 42 studies, Adderley et al. demonstrated that the pooled estimate for the BALF lymphocyte percentage in chronic HP was 43% compared to 10% of lymphocytes in IPF [25]. Importantly, a similar pooled estimate of 44% for BALF lymphocytes was provided when the sensitivity analysis using 26 studies that defined chronic HP based on signs of fibrosis on HRCT and/or lung biopsy. Additionally, the authors chose to use individual patient data from eight studies, pooled and analysed as a single cohort, to calculate the performance characteristics of BALF at different lymphocyte percentage thresholds to discriminate chronic HP from IPF/non-IPF idiopathic interstitial pneumonia (IIP). The thresholds that maximised sensitivity and specificity were 20% and 50%, respectively. If the cut-off level of BALF lymphocyte percentage was set at 20%, the positive predictive value was only 57% for suggesting chronic HP vs. IPF/non-IPF IIP. On the other hand, if the cut-off level was set high at 50%, fewer subjects with chronic HP would be captured but with minimising false-positives (PPV increased to 78%). The BALF lymphocytosis value that concurrently optimised sensitivity (70.7%) and specificity (67.6%) was 21% [25]. Similarly, in a meta-analysis of 36 studies used to inform the ATS/JRS/ALAT guidelines specifically, Patolia et al. reported comparable sensitivity (69%) and specificity (61%) for a lymphocytosis threshold of 20% distinguishing fHP from IPF [26]. However, despite a relatively high mean difference of 21% in the BALF lymphocyte percentage (95% confidence interval, 14–27%) of fHP versus IPF patients, the area under the receiver-operating-characteristic curve (AUC) was only 0.54. After checking the results the suggested primary explanation was the high standard deviation (SD) observed for the BALF lymphocyte percentage in both fHP and IPF populations within many studies. A threshold of 40% greatly increased specificity to 93%, but markedly reduced sensitivity to 41%, whereas a threshold of 30% gave a specificity of 80% and the sensitivity of 55% in distinguishing fHP from IPF [26]. In our study, the mean difference in the BALF lymphocyte percentage between fHP and IPF was slightly higher (25%), and the AUC value was excellent (0.90) for lymphocytosis and fair (0.71) for total cell count. 

The ATS/JRS/ALAT guideline committee did not identify a threshold proportion of BALF lymphocytes that distinguishes HP from non-HP ILD, given the poor area under the curve for the comparisons. Based on the committee’s collective clinical experience, the 30% threshold for lymphocytosis in BALF has been considered to be reasonable [2]. In contrast, in a recent Delphi online survey involving 45 ILD experts from 14 countries, the vast majority of experts rated BALF lymphocytosis more than 40% as “important” for the diagnosis of chronic HP, while levels between 30 and 39% were considered a grey zone (did not meet consensus), and values between 20 and 29% as unhelpful [27]. 

In turn, the absence of BALF lymphocytosis supports the diagnosis of IPF [28,29]. Ohshimo et al. were the first to demonstrate that the cut off level of <30% for lymphocytes in BALF had a favourable discriminative power for the diagnosis of IPF [30]. 

Tzilas et al. re-evaluated the initial diagnoses of patients with fibrotic ILD and indeterminate for UIP HRCT pattern during multidisciplinary team discussion. The authors reported that a BALF lymphocytosis of ≥20% was of added diagnostic value in their retrospective cohort of undiagnosed fibrotic ILD, changing the diagnosis from IPF to HP in 15% of the cases [11]. They also stated that even a mild BALF lymphocytosis (>20–25%) should increase vigilance for the search of an underlying inciting antigen. 

In our group, the percentage of lymphocytes in the BALF did not exceed the threshold of 30% in any of the patients with IPF, and only in 10% of the patients with IPF was the lymphocytosis in the BALF higher than 20%. On the other hand, among patients with fHP, the threshold of 30%, considered reasonable by the ATS/JRS/ALAT guidelines committee, was exceeded by 60% of patients, while the threshold of 40% proposed by the DELPHI survey was exceeded by only 42% of patients [2,27].

Very little data exist on the total cell count in BAL fluid. Domagala-Kulawik et al. reported that elevated total cell count in the BALF is common in patients with ILD [31]. According to the authors, the risk of HP diagnosis increased by 37% per one million of BALF cells compared to a control group of healthy volunteers. Bergantini et al. showed that the total cell count was significantly higher in the fibrotic HP group than in the connective tissue disease (CTD)-ILD and cryptogenic organizing pneumonia (COP) groups [32]. However, we have shown that both the total cell count and the percentage of lymphocytes were significantly higher in the fHP group compared to the IPF group. The probability of fHP diagnosis increased in our study by 4% per one million of BALF cells. Including in the diagnostic process of fibrotic HP not only the percentage of lymphocytes, but also the total number of cells could be an additional factor supporting the diagnosis. 

The above-mentioned systematic reviews and meta-analyses included only observational studies reporting on the BALF lymphocyte percentage in patients with HP, as no randomised controlled trials or controlled observational studies exist on this topic [25,26]. What is important is that prior studies assessing the diagnostic utility of BALF lymphocytosis in HP may be limited by varied diagnostic criteria. Additional confounding may also involve the unclear differentiation of fibrotic vs. nonfibrotic HP with the use of prior classification schemes (acute, subacute, and chronic), where BALF lymphocytosis findings may intrinsically vary. Our results are derived from a cohort of well-characterized fHP and IPF patients, in which the diagnosis was re-evaluated according to the latest international guidelines [2,6,12]. 

BALF total and differential cell count may be influenced by several confounding factors such as age, smoking status, systemic corticosteroid therapy, the presence and extent of fibrosis and duration of symptoms [8,14,25,31]. According to the Domagala-Kulawik et al., cigarette smoking caused approximately two-fold increase in the BALF total cell count (TCC) when healthy smokers and non-smokers have been compared [31]. Additionally, the authors observed the higher mean TCC values in ILD patients than in healthy smokers. Adderley et al., using individual patient data from eight studies, showed that older age and ever smoking have been associated with lower BALF lymphocyte percentages in patients with chronic HP [25]. On the other hand, an increase in the TCC with a significant elevation in the BALF lymphocyte percentage (>30%) is considered to play a key role in distinguishing fHP from IPF [2,10]. Hence, some authors have suggested using the threshold of 20% for BALF lymphocytosis in smoking HP patients [33]. In our study, the patients with fHP were significantly younger, more often female and never-smokers in comparison to IPF patients. Given that age and smoking were major potential confounding factors, we used them as covariates in our multivariable analyses, thus eliminating or minimising any bias that they may have had on our results. Additionally, the sensitivity analysis using the multivariable logistic regression models further adjusted for estimated pack-years of smoking, provided similar findings (Appendix A).

Our study has some limitations. First, an obvious limitation is the retrospective, single-centre nature of the study. On the other hand, BAL is a procedure performed in our centre by experienced staff for several decades, always according to international guidelines. Hence, the impact of the BAL methodology on the results, as may be seen in a multicentre study, was limited. Second, a relatively small number of patients were included in the study because we focused on IPF and fHP patients who underwent a BAL procedure. Finally, an essential limitation of the study was incorporation bias, since the presence of BALF lymphocytosis was considered one of the criteria to diagnose fHP. However, BALF lymphocytosis was not the only test to confirm or rule out the HP diagnosis. A significant proportion of our patients with fHP had TBLB performed simultaneously with BAL and subsequently, in doubtful cases, cryobiopsy or surgical lung biopsy, based on which the diagnosis could be established without taking the BAL results into account.

## 5. Conclusions

The percentage of lymphocytes and the total cell count in the BALF are important in separating fHP from IPF. Increased cellularity with lymphocytosis in BALF persists despite lung fibrosis in HP and may be an additional value in improving the diagnostic likelihood of fHP and allowing us to avoid more invasive tests such as lung biopsy. A BALF lymphocytosis exceeding the threshold of 20% increases the probability of fHP diagnosis by twenty-five times compared to IPF. Therefore, even a mild BALF lymphocytosis should increase diagnostic vigilance and call attention to a more thorough search for a causative antigen or history of exposure.

## Figures and Tables

**Figure 1 diagnostics-13-00935-f001:**
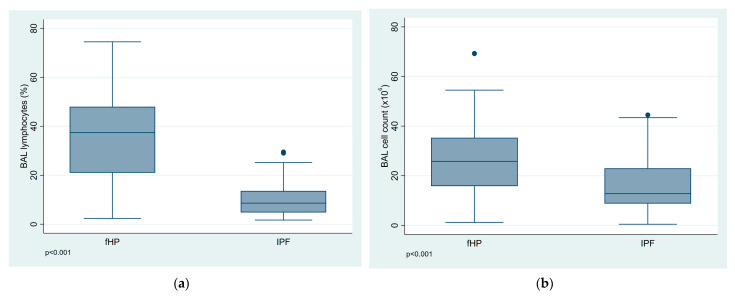
Bronchoalveolar lavage fluid results in relation to diagnosis fibrotic hypersensitivity pneumonitis (fHP) and idiopathic pulmonary fibrosis (IPF). (**a**) The percentage of BALF lymphocytes and (**b**) the BALF total cell count in both groups.

**Figure 2 diagnostics-13-00935-f002:**
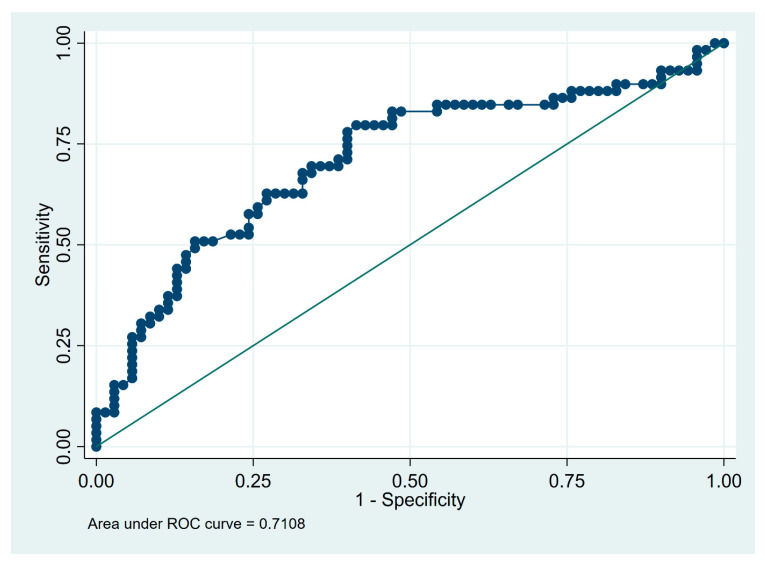
Receiver operating characteristic (ROC) curve for diagnostic utility of total cell count in differentiating between fHP (*n* = 65) and IPF (*n* = 71). Dots indicate individual data points used to construct the graph and calculate the value of the area under the ROC curve.

**Figure 3 diagnostics-13-00935-f003:**
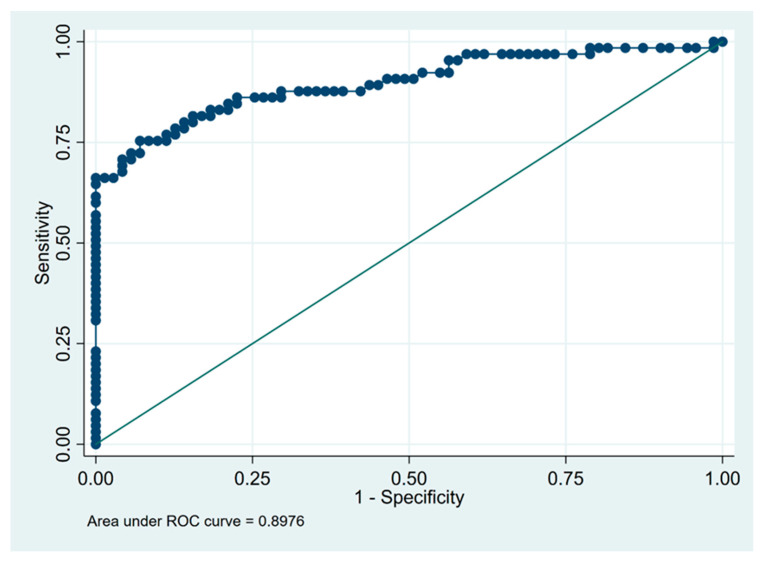
Receiver operating characteristic (ROC) curve for diagnostic utility of bronchoalveolar lavage fluid lymphocytosis in differentiating between fHP (*n* = 65) and IPF (*n* = 71). Dots indicate individual data points used to construct the graph and calculate the value of the area under the ROC curve.

**Table 1 diagnostics-13-00935-t001:** Baseline characteristics of the patients with fibrotic hypersensitivity pneumonitis and idiopathic pulmonary fibrosis.

Parameter	fHP	IPF	*p*-Value
**Subjects**	65	71	
**Age at diagnosis, years**	54.97 (10.87)	64.00 (7.18)	<0.001
**Sex**			0.012
**Male**	32 (49.2)	50 (70.4)	
**Female**	33 (50.8)	21 (29.6)	
**Smoking status**			<0.001
**Smokers and ex-smokers**	26 (40.0)	55 (77.5)	
**Never-smokers**	39 (60.0)	16 (22.5)	
**Pack-years**	8.75 (16.68)	35.07 (22.08)	<0.001
**Identified exposure to:**	47 (72.3)	8 (11.3)	<0.001
**Poultry**	14	2	
**Pigeons**	16	1	
**Parrots**	4	4	
**Hay/feed**	15	1	
**Other**	10	0	

Data are presented as *n*, mean ± SD or *n* (%), unless otherwise stated, *p*-values were calculated using one-way ANOVA and chi-squared test; fHP: fibrotic hypersensitivity pneumonitis; IPF: idiopathic pulmonary fibrosis.

**Table 2 diagnostics-13-00935-t002:** Baseline pulmonary function test and echo results of the study population.

Parameter	fHP *n* = 65	IPF *n* = 71	*p*-Value
FVC, % predicted	77.85 (19.29)	87.14 (17.69)	0.004
FEV1, %predicted	73.92 (19.12)	89.55 (17.02)	<0.001
FEV1/FVC	80.66 (8.36)	80.80 (8.85)	0.924
TLC, %predicted	79.67 (15.81)	84.56 (16.85)	0.088
TL,co, %predicted	47.20 (16.25)	52.29 (16.00)	0.071
6MWD, m	496.00 (92.67)	465.94 (111.12)	0.092
Desaturation during 6MWT, %	7.83 (5.91)	6.38 (5.89)	0.157
TVPG, mmHg	31.02 (11.22)	31.45 (7.72)	0.815

Data are presented as mean ± SD, unless otherwise stated, *p*-values were calculated using one-way ANOVA; fHP: fibrotic hypersensitivity pneumonitis; IPF: idiopathic pulmonary fibrosis; FVC: forced vital capacity, FEV1: forced expiratory volume in one second, FEV1/FVC: forced expiratory volume in one second to forced vital capacity ratio; TLC: total lung capacity, TL,co: transfer factor of the lungs for carbon monoxide, TVPG: tricuspid valve transvalvular pressure gradient; 6MWD: 6-min Walk Distance; 6MWT: 6-min Walk Test.

**Table 3 diagnostics-13-00935-t003:** Bronchoalveolar lavage fluid percent differential and lymphocytosis distribution in the study group.

Parameter	fHP *n* = 65	IPF *n* = 71	*p*-Value
Total cell count (×10^6^/mL)	26.23 (15.25)	16.20 (10.38)	<0.001
Lymphocytes (%)	35.24 (17.50)	10.17 (6.80)	<0.001
Eosinophils (%)	3.11 (3.68)	3.81 (4.30)	0.323
Neutrophils (%)	7.73 (6.93)	6.74 (6.11)	0.379
Macrophages (%)	54.51 (17.47)	79.28 (10.01)	<0.001
Lymphocytosis ≥ 20%, *N* (%)	49 (75.4)	7 (9.9)	<0.001
BALF lymphocytosis, *N* (%)			
<10%	8 (12)	44 (62)	-
10–20%	8 (12)	21 (30)	-
21–30%	10 (15)	6 (8)	-
31–40%	12 (18)	0	-
41–50%	15 (23)	0	-
51–60%	7 (11)	0	-
>60%	5 (8)	0	-

Data are presented as mean ± SD or *N* (%), unless otherwise stated, *p*-values were calculated using one-way ANOVA and chi-squared test; BALF: bronchoalveolar lavage fluid; fHP: fibrotic hypersensitivity pneumonitis; IPF: idiopathic pulmonary fibrosis.

**Table 4 diagnostics-13-00935-t004:** Clinical predictors of fibrotic hypersensitivity pneumonitis vs. idiopathic pulmonary fibrosis diagnosis (logistic regression analysis ^#^).

Characteristics	OR	95% CI	*p*-Value
**Male**	0.90	0.36–2.22	0.814
**Age at diagnosis**	0.90	0.86–0.94	<0.001
**Ever smoker**	0.20	0.08–0.49	<0.001
**Identified exposure**	17.21	5.93–50.00	<0.001
**FVC, % predicted**	0.99	0.97–1.01	0.424
**FEV1, %predicted**	0.97	0.95–0.99	0.018
**FEV1/FVC**	0.97	0.93–1.02	0.239
**TLC, %predicted**	0.99	0.96–1.02	0.423
**TL,co, %predicted**	0.97	0.95–1.00	0.068
**6MWD, m**	1.00	1.00–1.00	0.885
**Desaturation during 6MWT, %**	1.04	0.97–1.12	0.237
**Total cell count in BALF**	1.04	1.01–1.08	0.020
**Neutrophils in BALF**	1.03	0.97–1.10	0.340
**Eosinophils in BALF**	0.98	0.89–1.08	0.692
**Lymphocytes in BALF**	1.16	1.09–1.23	<0.001
**Lymphocytosis in BALF > 20%**	25.06	7.43–84.50	<0.001

^#^ Adjusted for a priori age, sex and smoking history. BALF: bronchoalveolar lavage fluid; FVC: forced vital capacity, FEV1: forced expiratory volume in one second, FEV1/FVC: forced expiratory volume in one second to forced vital capacity ratio; TLC: total lung capacity, TL,co: transfer factor of the lungs for carbon monoxide; 6MWD: 6-min Walk Distance; 6MWT: 6-min Walk Test.

**Table 5 diagnostics-13-00935-t005:** Optimal cut-off values of lymphocytosis and total cell count in BALF for differentiation of fibrotic hypersensitivity pneumonitis from idiopathic pulmonary fibrosis.

Parameter	Cut-off	Specificity	Sensitivity	PPV	NPV	AUC
**Lymphocytosis in BALF (%)**	21	0.93	0.75	0.91	0.80	0.84
**Total cell count in BALF (×10^6^)**	15	0.59	0.80	0.62	0.77	0.69

BALF: bronchoalveolar lavage fluid, PPV: positive predictive value; NPV: negative predictive value; AUC: area under the receiver operating characteristic (ROC) curve.

## Data Availability

The data are available from the corresponding author upon request.

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
