# Peer review of "Bronchoalveolar Lavage Cell Count and Lymphocytosis Are the Important Discriminators between Fibrotic Hypersensitivity Pneumonitis and Idiopathic Pulmonary Fibrosis"

_diagnostics, 2023, doi:10.3390/diagnostics13050935_

Round 1

Reviewer 1 Report

more explain in the discussion about the risk factors affected in both FHP and IPF cases as smoking status.

clear the abbreviations under the tables as the FEV1/FVC in table 4 

Reviewer 2 Report

1)      How currently the patients are discriminated between fibrotic hypersensitivity pneumonitis and idiopathic pulmonary fibrosis in clinics?

2)      Can CT scan substitute for bronchoscopy for this discrimination? Since bronchoscopy is very invasive. Moreover, this method is not applicable as a routine diagnostic method.

3)      Is there bronchoscopy data deposited in a public repository?

4)      Why the errors bars in Fig 1 are very large?

5)  Figures 2,3 does not have a legend enough to describe the figures. Figures should be self-informative.  

Reviewer 3 Report

In this retrospective cohort study, the authors aimed to determine the value of bronchoalveolar lavage (BAL) total cell count (TCC) and lymphocytosis in distinguishing fibrotic HP and IPF and to evaluate the best cut-off points discriminating these two fibrotic ILD. After comprehensive analyses, the authors provided the optimal cut-off values to differentiate fibrotic HP from IPF were 15x10^6 for TCC and 21% for BAL lymphocytosis with AUC 0.69 and 0.84, respectively. Overall the study is scientifically sound and the conclusion is reasonable. However, the major concern that I have with the manuscript is the baseline characteristics of the patients for the two fibrotic ILD are so significantly different in terms of age, smoking status, pack years, etc. How did the authors rule out these factors in drawing their conclusion? They should discuss more regarding the impact of these factors to the study. 

Round 2

Reviewer 2 Report

The authors provided the answers to my comment. The main issue that has been left is " bronchoscopy data must be deposited in a public repository." Or the authors must agree that they make access to data upon request. 
